# Experiences and perceptions of patients with ankylosing spondylitis: A systematic review and meta-synthesis of qualitative studies

**Yu Li , Dongchi Ma, Lili Yang ***

School of Nursing, Zhejiang Chinese Medical University, Hangzhou, Zhejiang Province, China

* yanglili@zcmu.edu.cn

## Abstract

### Objectives

The systematic evaluation of relevant qualitative studies on the experiences of patients with ankylosing spondylitis provides a foundation for the clinical development of personalized disease management programs for this patient category.

### Methods

Multiple databases, including PubMed, Web of Science, Embase, Cochrane Library, CINAHL, Scopus, CNKI, Wanfang, CBM, and VIP, were searched for qualitative research literature on the experiences of patients with ankylosing spondylitis from the inception of databases to March 2024. Eligible studies were included, and quality was assessed using the quality evaluation standard of qualitative research at the Joanna Briggs Institute (JBI), Australia (2016). The results were integrated using the meta-aggregation approach.

### Results

A total of 11 papers were included in the review. Four themes and 10 subthemes were synthesized: (1) difficulties in diagnosis and treatment; (2) effects of disease symptoms; (3) maladjustment of social roles; and (4) lack of support for disease response.

### Conclusion

Medical staff should leverage the advantages of the Internet to enhance knowledge and education on ankylosing spondylitis. They should focus on patients' mental health, assist in active self-management, provide personalized interventions, promote recovery, and improve the quality of life for patients. Additionally, society should offer a diverse range of support.

**Data Availability Statement:** All relevant data are within the manuscript and its Supporting information files.

**Funding:** The author(s) received no specific funding for this work.

**Competing interests:** The authors have declared that no competing interests exist.

## Introduction

Spondyloarthritis (SpA) is a general term for inflammatory immune-mediated diseases, which can be categorized into axial spondyloarthritis (axSpA) and peripheral spondyloarthritis (pSpA) based on their manifestations [1]. AxSpA primarily involves the sacroiliac joints and spine, including non-radiographic axial SpA (nr-axSpA) and ankylosing spondylitis (AS). PSpA is characterized by peripheral arthritis, enthesitis, and dactylitis, and includes psoriatic arthritis, reactive arthritis, and arthritis associated with inflammatory bowel disease (IBD) [2–4].

AS, a type of axSpA, is characterized by significant structural changes in the sacroiliac joints visible on radiographs [5]. The disease is primarily influenced by genetic and environmental factors. Current research indicates that both autoinflammatory and autoimmune factors may play consecutive roles in the pathogenesis of AS, with the concept of an autoimmune response involving specific autoantibodies gaining traction. AS prevalence varies globally, being highest in North America, followed by Europe, Asia, Latin America, and Africa, with a male-to-female ratio of approximately 3:1 [5, 6]. However, recent studies suggest that misdiagnosis of inflammatory back pain, variability in the global distribution of human leukocyte antigen B27 (HLA-B27), and cultural biases contribute to the underdiagnosis of AS in women [7].

AS is characterized by inflammatory damage to the mid-axis skeleton, including the spine, sacroiliac joints, and spinal attachment points, leading to progressive loss of joint function, chronic back pain, spinal dyskinesia, and extra-skeletal organ complications [8, 9]. Patients typically experience low back pain of varying intensity at night and morning stiffness, which usually subsides with activity. Since patients are generally under 45 years of age and in the prime of their careers, chronic low back pain significantly impacts their work and creates a financial burden [10]. One major reason for the extended diagnosis delay (DD) compared to other rheumatic diseases is that the early stages of AS are challenging to detect using X-rays [11, 12]. AS progresses slowly, has a long disease duration, and has a high disability rate.

Through a systematic review of the literature, researchers have determined that TNF inhibitors (TNFi), IL-17 inhibitors (IL-17i), JAK inhibitors (JAKi), and nonsteroidal anti-inflammatory drugs (NSAIDs) are efficacious and safe for treating ankylosing spondylitis. Health education and exercise programs have also proven effective [13, 14]. Additionally, traditional Chinese therapies such as cupping, moxibustion, and turmeric extract have emerged as significant research trends in recent years [15–18]. Despite recent advancements in treatments, there is no cure, requiring patients to manage the disease for many years. This chronic condition often leads to social isolation, increased negativity, and decreased quality of life [19]. Studies have shown that depression and anxiety are the most common psychological characteristics of AS patients [20], and AS may also be associated with the development of schizophrenia (SCZ) and anorexia nervosa (AN) [21].

Focusing on patients' experiences and perspectives on illness is critically important in modern healthcare [22]. Patients' experiences and perspectives enable healthcare providers to fully understand the impact of disease on patients' quality of life, allowing for the development of more personalized treatment plans. Understanding patients' experiences of illness can help identify deficiencies in the healthcare process, promote improvements and innovations in healthcare services, enhance the overall quality of healthcare, and contribute to the advancement of the medical field [23]. Therefore, emphasizing patients' experiences and perspectives is not only a sign of respect and care for the patients themselves but also holds significant importance in improving medical outcomes and fostering medical progress [24].

Current research on AS is predominantly quantitative. In recent years, there has been growing interest in the experiences of AS patients in various countries. However, a single qualitative

study is often insufficient to provide effective guidance for clinical practice or patient self-management [25]. Consequently, this study employs a pooled and integrated approach to systematically elucidate the inner experiences of AS patients. This approach provides a foundation for clinical staff to develop coping strategies and construct personalized patient care management plans.

## Materials and methods

### Protocol registration and reporting

The review protocol was registered in the International Prospective Register of Systematic Reviews (PROSPERO) database (CRD42024548012). We adhered to the Preferred Reporting Items for Systematic Reviews and Meta-Analyses (PRISMA) [26] guidelines and the Enhancing Transparency in Reporting the Synthesis of Qualitative Research (ENTREQ) [27] statement.

### Inclusion criteria

The inclusion criteria for this study followed the PICoS principle. Population: patients diagnosed with AS. Interest of phenomena: the experiences and feelings of AS patients. Context: the environments in which patients were treated, lived, and worked after being diagnosed with AS. Study design: all types of qualitative research, including but not limited to phenomenology, grounded theory, narrative research, ethnography, etc.

### Data sources and search strategy

Qualitative research on the experiences and perceptions of AS patients was comprehensively searched in the databases of PubMed, Web of Science, Embase, Cochrane Library, CINAHL, Scopus, CNKI, Wanfang, CBM, and VIP. To ensure the inclusion of all relevant literature, the references of the included articles were also traced. The details of our search strategy are presented in S1 Appendix.

### Study selection

Two researchers who had received training in qualitative research and evidence-based nursing courses independently screened the literature, extracted the data, and then cross-checked. If the opinions could not be agreed upon, the third researcher was asked to assist in the determination. Literature screening steps: import the literature into EndNote X9; eliminate duplicate literature; according to the inclusion and exclusion criteria; read the title and abstract; and exclude the literature not related to the topic.

### Assessment of methodological quality

Two researchers independently evaluated the quality of the included literature according to the Joanna Briggs Institute Library (JBI) evidence-based health care center qualitative research quality evaluation standard (2016 edition). After completing the evaluation, the results were compared, and a third researcher assisted in resolving any discrepancies. This qualitative assessment consists of 10 items, each evaluated as 'yes,' 'no,' or 'unclear.' The evaluation results are quantified, with 'yes' scoring 1 point, and 'no' or 'unclear' scoring 0 points, resulting in a total possible score of 10 points.

### Data extraction

Two researchers independently extracted the information using the JBI general information extraction form. The extraction primarily included the first author's name, year of publication, country, research method, data collection method, study population, phenomenon of interest, and main findings. In cases of discrepancies, a third researcher made the final decision.

### Data synthesis

The data is synthesized through the JBI aggregation method in three steps. In the first phase, all results from the included studies were extracted with descriptions, and the credibility of the results for each study was assessed. In the second phase, categories were developed for the extracted results, with at least two findings per category. In the third phase, these categories were synthesized into a new set of combined results. The use of meta-aggregation allows for a synthesis of the experiences and feelings of people with ankylosing spondylitis, independent of bias, ego, or external factors.

### Credibility assessment of integration results

The confidence level of the integration results was evaluated using the ConQual approach [28]. Based on the reliability and confidence assessments, the quality level of the body of evidence was determined, indicating whether the quality was reduced or maintained.

## Results

### Study selection

As shown in Fig 1, the initial search yielded a total of 624 papers. After eliminating duplicates using EndNote X9, 577 papers remained. From these, 36 papers were initially screened based on titles and abstracts, and 11 papers were retained after full-text reading, re-screening, and quality evaluation. These studies, published between 2008 and 2022, were conducted in seven countries: China, the United States, Iran, Denmark, Turkey, the United Kingdom, and Norway. All studies were qualitative in nature. A total of 160 patients were interviewed, including 103 men and 57 women [29–39]. Study characteristics are reported in Table 1.

### Methodological quality of studies

The results of the quality appraisal of included studies are presented in Table 2. A total of 11 papers were included, all scoring between 7 and 10. The primary issue was that the studies did not describe the theoretical background of the researchers or elaborate on the interplay between the researchers and the studies.

### Confidence in the findings

ConQual was used to assess the confidence of the integration results. Of the four integrated results, one theme was rated as having high confidence, while three themes were rated as having moderate confidence. As shown in Table 3.

### Main findings

In the 11 studies, we extracted a total of 40 findings, classified and grouped them by similar meanings, ultimately forming 10 subthemes and integrating 4 themes. As shown in Table 4.

**Difficulties in diagnosis and treatment.** *Medical experience twists and turns.* AS is characterized by an insidious onset, with early symptoms that are not obvious enough for

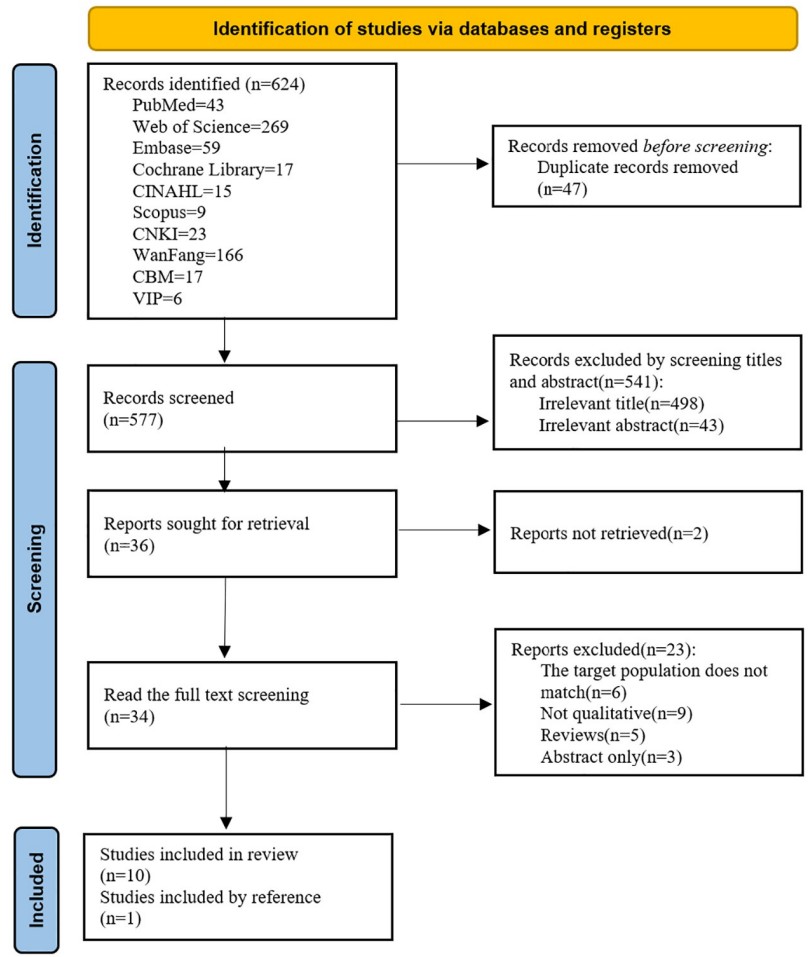

**Fig 1. PRISMA flowchart of initial searches and inclusion.**

detection. Patients in the early stages often ignore minor symptoms [38]. These insufficiently obvious symptoms can lead to diagnostic difficulties, delays, and even misdiagnosis by doctors [36, 39]. Additionally, patients face multiple referrals, which significantly drain their time and energy [39].

*The dilemma of pain medication use.* Taking pain medication can reduce pain and help patients maintain a normal life, but the side effects often reduce compliance. Patients need medication to accomplish work tasks and improve sleep quality in their daily lives [32, 37]. However, the side effects of the medication are a cause for concern [31]. Moreover, online information about medication use is inconsistent, making it difficult for patients to recognize their effectiveness [39].

**Effects of disease symptoms.** *The diagnosis of the disease resulted in a negative emotional state.* Patients typically experience numerous negative emotions after being diagnosed with the disease, including confusion, fear, pessimism, helplessness, frustration, and even depression. Some patients experience confusion about their future, facing difficulties in finding a partner and securing employment, and feeling reluctant to remain dependent on their parents. Upon learning that the disease is hereditary, they become concerned about its impact on the next generation [39]. Some patients become emotionally unstable, start to become irritable, and some even reach moderately severe depression [29, 36].

**Table 1. Study characteristic.**

| Author (year) | Country | Study design | Data collection method | Participants | Phenomenon of interest | Main findings |
|---|---|---|---|---|---|---|
| **Bagcivan et al. (2016)** [29] | Turkey | Phenomenological research | Semi-structured interviews | 23 people: 16 males 7 females | AS patients' experiences with pain and its effect on their lives. | Two themes were identified: (1) physical and social limitations. (2) emotional problems. |
| **Chakravarty et al. (2021)** [30] | US | Qualitative descriptive study | semi-structured group discussions | 12 people: 6 males 6 females | Identify the content of reported outcome indicators that best resonate with the patient experience. | Two themes were identified: (1) activities of daily living. (2) fatigue. |
| **Davies et al. (2013)** [31] | UK | Qualitative descriptive study | Focus group interviews | 14 people: 7 males 7 females | From the patients' perspective (i) the effect of fatigue in AS with a focus on (ii) ways of self-managing fatigue as part of their every-day lives, and (iii) to identify potential interventions for future research. | Three themes were identified: (1) pervasive fatigue. (2) current limitations (of self-management). (3) a new direction (for future interventions). |
| **Farren et al. (2021)** [32] | UK | Phenomenological research | Seven-day diary and semi-structured interview | 10 people: 6 males 4 females | The perceived causes, consequences and management of fatigue in AS. | Three themes were identified: (1) patterns and causes of fatigue. (2) consequences of fatigue. (3) management of fatigue. |
| **Madsen et al. (2014)** [33] | Denmark | Qualitative descriptive study | Semi-structured interviews | 13 males | How men experience AS and the challenges related to living with AS as a chronic disease. | Four themes were identified: (1) approaching a diagnosis. (2) ill in a social context. (3) challenged as a man. (4) the importance of remaining physically well. |
| **Mengshoel et al. (2010)** [34] | Norway | Qualitative descriptive study | Semi-structured interviews | 12 people: 4 males 8 females | The nature of fatigue and how it is managed in daily life situations by individuals with ankylosing spondylitis. | Two themes were identified: (1) comprehensible, manageable, life strain-related tiredness. (2) unfamiliar and unmanageable illness-related fatigue. |
| **Mengshoel et al. (2008)** [35] | Norway | Qualitative descriptive study | In-depth individual interviews | 12 people: 4 males 8 females | The relationship between illness fluctuations and how people with AS adapt to everyday life situations. | Three themes were identified: (1) ordinary life condition. (2) slowed-down life condition. (3) disrupted life condition. |
| **Primholdt et al. (2017)** [36] | Denmark | Phenomenological research | Semi-structured interviews | 5 males | How patients live from day to day, and how they meet the challenges they face in relation to their disease. | Three themes were identified: (1) daily living and psychological reactions. (2) a difficult diagnosis. (3) working life and identity. |
| **Shahba et al. (2021)** [37] | Iran | Phenomenological research | Semi-structured interviews | 28 people: 25 males 3 females | The life experience of Iranian AS patients. | Three themes were identified: (1) always with pain. (2) the perceived limitation. (3) fearing the unknown future. |
| **Zhu et al. (2022)** [38] | China | Phenomenological research | Semi-structured in-depth individual interviews | 16 people: 9 males 7 females | The disease perception and experience of young and middle-aged patients with ankylosing spondylitis. | Four themes were identified: (1) impact of disease symptoms. (2) limitations in life planning. (3) experience of loneliness and isolation. (4) multiple strategies for coping with disease. |

*(Continued)*

**Table 1.** (Continued)

| Author (year) | Country | Study design | Data collection method | Participants | Phenomenon of interest | Main findings |
|---|---|---|---|---|---|---|
| **Zhu et al. (2022)** [39] | China | Phenomenological research | Semi-structured in-depth individual interviews | 15 people: 8 males 7 females | Characteristics of disease perception at different stages in young patients with ankylosing spondylitis. | Eleven themes were identified: perception of abnormal somatic symptoms prior to diagnosis; tortuous experience of seeking medical treatment; prominent perception of adverse emotions after diagnosis; multiple ways of seeking information and support for the disease; perceived benefits of treatment; overly optimistic prognosis; ignoring the risk of progression of the disease during the period of recovery; skepticism about the controllability of the treatment; active or passive acceptance of the disease; declining level of social participation; desire for understanding and support. |

*Pain brings challenges to daily life.* The disease significantly limits patients' daily life and work due to chronic pain. During painful episodes, patients may be unable to use the toilet independently or walk normally [29, 35]. Prolonged pain reduces sleep quality, lowers tolerance, and affects sexual life [29, 31, 37]. Some patients even lose their jobs due to the pain [36].

*Fatigue affects the quality of life.* Fatigue caused by the disease can lead to physical discomfort and affect the quality of life. The fatigue that accompanies the disease is usually chronic,

**Table 2. Results of the critical appraisal of the studies included.**

| | Q1 | Q2 | Q3 | Q4 | Q5 | Q6 | Q7 | Q8 | Q9 | Q10 | Total |
|---|---|---|---|---|---|---|---|---|---|---|---|
| Bagcivan et al. (2016) [29] | Y | Y | Y | Y | Y | Y | U | Y | Y | Y | 9 |
| Chakravarty et al. (2021) [30] | Y | Y | Y | Y | Y | N | N | Y | N | Y | 7 |
| Davies et al. (2013) [31] | Y | Y | Y | Y | Y | N | Y | Y | Y | Y | 9 |
| Farren et al. (2021) [32] | Y | Y | Y | Y | Y | N | Y | Y | Y | Y | 9 |
| Madsen et al. (2014) [33] | Y | Y | Y | Y | Y | Y | N | Y | Y | Y | 9 |
| Mengshoel et al. (2010) [34] | Y | Y | Y | Y | Y | N | Y | Y | Y | Y | 9 |
| Mengshoel et al. (2008) [35] | Y | Y | Y | Y | Y | N | Y | Y | Y | Y | 9 |
| Primholdt et al. (2017) [36] | Y | Y | Y | Y | Y | Y | Y | Y | Y | Y | 10 |
| Shahba et al. (2021) [37] | Y | Y | Y | Y | Y | Y | N | Y | Y | Y | 9 |
| Zhu et al. (2022) [38] | Y | Y | Y | Y | Y | N | N | Y | U | Y | 7 |
| Zhu et al. (2022) [39] | Y | Y | Y | Y | Y | N | N | Y | U | Y | 7 |

Q1. Is there congruity between the stated philosophical perspective and the research methodology?

Q2. Is there congruity between the research methodology and the research question or objectives?

Q3. Is there congruity between the research methodology and the methods used to collect data?

Q4. Is there congruity between the research methodology and the representation and analysis of data?

Q5. Is there congruity between the research methodology and the interpretation of results?

Q6. Is there a statement locating the researcher culturally or theoretically?

Q7. Is the influence of the researcher on the research, and vice- versa, addressed?

Q8. Are participants, and their voices, adequately represented?

Q9. Is the research ethical according to current criteria or, for recent studies, and is there evidence of ethical approval by an appropriate body?

Q10. Do the conclusions drawn in the research report flow from the analysis, or interpretation, of the data?

Y, yes; N, no; U, unclear.

**Table 3. ConQual summary of the findings.**

| Synthesised finding | Type of research | Dependability | Credibility | ConQual score |
| --- | --- | --- | --- | --- |
| **Difficulties in diagnosis and treatment** | Qualitative | — | Downgrade 1 level | Moderate |
| **Effects of disease symptoms** | Qualitative | — | — | High |
| **Maladjustment of social roles** | Qualitative | — | Downgrade 1 level | Moderate |
| **Lack of support for disease response** | Qualitative | — | Downgrade 1 level | Moderate |

and this fatigue can lead to discomfort in certain organs of the patient's body [30, 31]. Fatigue can also have some other effects, such as a lack of energy, decreased self-image management ability, memory loss, etc. [32, 35].

**Maladjustment of social roles.** *Weakened family functioning*. In the family system, the original roles and functions of family members are affected by the limitations of the disease. Male patients often struggle to accept unemployment, influenced by the education they received in childhood [33]. Unmarried female patients plan to expedite marriage and child-bearing [29]. Married women who have given birth are no longer interested in having more children [29]. The hereditary nature of the disease affects couples' willingness to have children [38].

*Career planning was disrupted*. The physical limitations imposed by the disease often disrupt patients' original career plans and reduce their occupational choices. Some patients are unable to continue their previous work [38]. Due to a lack of physical endurance, some patients cannot handle long working hours or participate in full-time employment [35, 36].

*There is social isolation*. The emergence of social isolation in patients is both passive and active. Patients actively avoid socializing with friends due to physical sensitivity and a desire to avoid contact with the outside world [29, 37]. When friends do not understand the disease, patients may be excluded from group activities [38]. In severe cases, the patient's original social connections may be severed [29, 35].

**Lack of support for disease response.** *Lack of emotional support*. When it comes to emotional support, patients typically receive insufficient assistance from various sources. For example, they experience a lack of understanding from colleagues, trust from parents, tolerance from friends, and support from partners [33, 36, 39].

*Lack of social support*. Patients face challenges such as employment and treatment without adequate social support to find solutions. When seeking help at an employment center, they often have to expend considerable effort explaining their situation to the staff [34]. Although hydrotherapy has a significant curative effect, its cost is high [31]. Additionally, some patients feel that mutual aid associations provide limited assistance [33].

## Discussion

In this study, it is evident that both healthcare professionals and the public have an insufficient understanding of AS. A lack of awareness among healthcare professionals hinders accurate disease diagnosis and may compromise the effectiveness of subsequent treatment [11]. Similarly, the public's unfamiliarity with AS can lead to misconceptions about patients, which negatively impacts patient responses to the disease and undermines social harmony. Therefore, it is important to enhance knowledge of the disease. Healthcare professionals can establish a dedicated group focused on AS health literacy to promote the concept that 'everyone is the first person responsible for their own health.' This initiative can leverage the internet along with print, screen, and mobile media to provide the public with information through texts, posters, and videos, thus enhancing the impact of disease education [40, 41]. Additionally, it is crucial

Table 4. Detailed thematic synthesis.

| Themes | Subthemes | All relevant quotes |
|---|---|---|
| **Difficulties in diagnosis and treatment** | Medical experience twists and turns | "It also hurt before, but it wasn't so serious, so I didn't pay attention to it. Now the attack is quite severe. Only if there is no pain for one or two days a week, I think of coming to the hospital again." [38]<br>The physicians could not explain Janus' pain attacks. It was not until a family member suggested that it could be AS that he had a magnetic resonance imaging (MRI) scan, was referred to a rheumatology department and soon after was diagnosed with AS–one year after the first episode of pain attacks. [36]<br>Despite the daily pain and a hard, physically demanding job, Mikkel managed to do this job. When his pain grew worse, he eventually went to a chiropractor to obtain pain relief. The chiropractor referred him for an MRI of the spine. Shortly afterwards, Mikkel was diagnosed with AS– 13 years after symptom onset. [36]<br>"The orthopedic (doctor) told me to see the pain department, the pain department (doctor) told me to see the orthopedic department, and then the Chinese medicine department, and none of them could see what was wrong with me." [39]<br>"The town doctor recommended minimally invasive surgery at an orthopedic clinic, and then the knee still hurt, and it took 7 or 8 years of visits before it was diagnosed." [39] |
| | The dilemma of pain medication use | "The highest level of pain is often in the mornings. I am not able to do my own responsibilities if I do not take my drugs as if living a normal life depends on medicines." [37]<br>"It works, and the benefit is that I sleep better at night." [32]<br>"I was taking indomethacin for the pain but it caused the bowel to flare up and I was hospitalised with that. . ..so I try and manage without drugs if I can. . .but of course you have to if the pain is very bad." [31]<br>"Some people on the internet say it's better to use oral medication, others say it's better (to use) biologics, and I don't know which (claim) to believe." [39] |
| **Effects of disease symptoms** | The diagnosis of the disease caused a bad mood | "I'm only in my thirties, I can't find a wife, I can't find a job, do I have to rely on my parents?" [39]<br>"The doctor said it's a hereditary disease. I'm afraid it'll affect the next generation." [39]<br>"How could a living, breathing person get this disease? I wanted to jump from the hospital." [39]<br>"I am frustrated that I have no stamina, that I'm such a weakling. That I can't just suck it up and then stay at work, and that I have to go home early." [36]<br>"Well, I was unemployed and became depressed. It wasn't serious, but my mood was like I wasn't sad or anything, but I quickly became angry and upset, and if I talked to my family or something, I would snap all the time, and that wasn't nice. I was sweating a lot and my heart pounded when I was going to sleep, so I couldn't sleep and all that kind of stuff. So I was given antidepressants to take, which wasn't nice–but that's how it is." [36]<br>"What about the depression? I had mild to moderate depression, said my doctor. Not true, said the psychologist. It is moderate to severe depression that you have, and I actually think that you need some pills for it. I was already taking 16 pills daily." [36]<br>"I never want to remember the really painful periods I have had. How can it be described?. . . I cannot find words to describe this situation. Just think: you cannot even put on your pants. I have never felt so helpless before." [29] |
| | Pain brings challenges to daily life | "My sex life has been very affected. Because of the very severe pain, I cannot have sex. I cannot adapt myself to sex because of the pain I feel. In fact, to lie down in bed, even for a very short time, increases my pain." [29]<br>"Going to the toilet was my biggest difficulty. You cannot sit on the toilet. If you can sit, you cannot stand up. This is a very bad situation. When it comes to cleaning yourself, you cannot bend or turn because of the pain." [29]<br>"I can't remember ever having a good night's sleep. I wake up constantly every couple of hours through pain, I never sleep through, so I am constantly tired." [31]<br>"For this disease, my nerves are on edge and with a chronic pain in my life, my tolerance is less than I used to be and sometimes, I feel like crying." [37]<br>"the days when my back hurts—what can I actually do then? I can't do anything." [36]<br>"It's not just bad, it's so bad I can't walk. It happened once at an airport where I had to use a wheelchair because I couldn't stand up. It's quite true, I couldn't stand up—my legs, or my leg, it gave way when I stood on it." [35] |
| | Fatigue affects the quality of life | "Fatigue and tiredness is something that I struggle with on a day-to-day basis. And in speaking with other rheumatologists in the past, it's something that can be underemphasized probably. So, I love that it's called out as one of the first questions there." [30]<br>"a burning sensation with your eyes when you are really tired." [31]<br>"'. . .I said, I can't eat it, I feel ill', and he said, 'what's wrong?', and I said, 'I have no energy, it's all gone, it makes me feel ill.'" [32]<br>"Because I'm tired most of the time, I can't be bothered putting on make-up or https://doing my hair." [32]<br>"I have to write it down because if I don't write it down it will get to three days later and I won't have remembered it at all." [32]<br>"I get noticeably more tired. It's like the feeling that I can't do anything. I'm a person who wants to do so much, I want to, I want to—but I don't manage to." [35] |
| **Maladjustment of social roles** | Weakened family functioning | "Even today, I feel bad about getting paid for just staying home. I don't contribute to anything. And it's because I was raised to believe a man gets up in the morning to be the breadwinner for his family." [33]<br>"In the future, there will be more pain affecting my life. For that reason, while my health is better, I should get married as soon as possible and have a child." [29]<br>"I do not think of a second pregnancy. My first pregnancy was a very difficult period. During and after my pregnancy, I had too much pain. I could not bend to pick up and nurse my baby. I do not want any more babies." [29]<br>"My wife and I were planning to have a second child, but then we found out about this disease, and I think I've heard that it has something to do with genetics, so I'd better forget about it." [38] |
| | Career planning was disrupted | "I now manage a shop that sells clothes, shoes, handbags and things. Generally, I'm very pleased with this. It's a fine job. But it wasn't what I intended to do. Because when I'd finished studying and got a job and worked very hard like young academics do, I had to go on sick leave. I had inflammation in my hip joints and back and everywhere. I was on sick leave for three months." [35]<br>"I am frustrated that I have no stamina, that I'm such a weakling. That I can't just suck it up and then stay at work, and that I have to go home early." [36]<br>". . .the days when my back hurts–what can I actually do then? I can't do anything–I definitely can't have a full-time job, so you can say that a lot of things are still unresolved." [36]<br>"To work as a customer service in a small county, although the salary is not high, the work is relatively loose and the body can bear it. During this period of time, the illness relapsed again. I have no choice but to quit." [38]<br>"I definitely can't have a full-time job, so you can say that a lot of things are still unresolved." [36] |
| | There is social isolation | "I do not want to go out with my friends. I become more sensitive when they put their hand on my joints and bones." [37]<br>"When you have pain, you do not think of your social life. You might want to think about it, but you cannot. . .In fact, I do not want to be with anybody during periods of pain." [29]<br>"Very difficult. For three months I could not get out of my house. I stopped going to the university. . .I was playing on the football team of my school, and now there is no football in my life." [29]<br>"I walked with a limp, so they (my friends) called me 'cripple' and stopped taking me to basketball games, which over time led to alienating myself from them." [38]<br>"I became a real patient—I must have been hospitalized about 70 or 80 times. I lost many of the friends I used to have. Yes, I did. First I lost contact with them for several years. And then you were very ill and things so you weren't the same when you came back." [35] |

*(Continued)*

**Table 4.** (Continued)

| Themes | Subthemes | All relevant quotes |
|---|---|---|
| **Lack of support for disease response** | Lack of emotional support | "When it's hard to go on a break, my coworkers might think I'm being hypocritical and uncomfortable when I'm not https://doing anything every day." [39]<br>"I hope the people around me can understand this disease, some things are not that I'm lazy and don't want to do, but I just can't do them." [39]<br>"...that I was retaining water in my hips, just like pregnant women. Then I was told that it was because I drank too much and partied too much-that it was because of that that I retained water in my hips." [36]<br>"One or two have dropped out because they couldn't really accept that you just couldn't go out with them every weekend." [33]<br>"...it is difficult for my partner to accept that I cannot manage usual tasks." [33] |
| | Lack of social support | "It's no wonder I'm so tired—I sleep so badly! And on top of everything it takes a lot of energy to explain the situation to the people at the job center who don't understand the sort of jobs I can do. Now I hope things will fall into place soon so I can use my energy to live the way I want to." [34]<br>"It would be lovely to swim in a warm pool but the prices these places are charging." [31]<br>The men spoke about patient associations negatively, although the majority of them had memberships. They did not take advantage of their offers, and they expressed no need to: '...sit in a circle and hear about someone else's pain.' [33] |

to address the health knowledge needs of remote areas by using traditional methods such as newspapers, publications, and books, and by actively organizing activities such as health clinics and health science education in villages. These efforts will enable more people to obtain, understand, and use information to maintain their health, thereby raising the overall health literacy of the population.

Psychological instability is a common problem in patients with AS, and interventions for psychiatric symptoms should be considered in treatment [42]. Healthcare professionals should provide timely dyadic coping interventions for diagnosed patients with partners to improve patients' psychological capital and reduce communication problems within the couple [43]. It is important to explain to patients and their families the significance of mental health in alleviating disease symptoms. They should be informed that persistent sadness, irritability, loss of interest in activities, sleep disorders, etc., require evaluation by a psychiatrist and active intervention for psychological issues [44]. Both AS and depression can manifest with pain and fatigue, making it crucial to identify depression through other symptoms to prevent missing the intervention period. Moreover, peer support programs can be beneficial during the early stages of coping following diagnosis, aiding patients in better adjusting to life with the disease [45]. Healthcare professionals can use social media to establish online communication platforms to help alleviate patient anxiety. The mHealth management platform should be introduced to AS patients who seek hospital consultations [46, 47]. Health education should also be provided to patients with limited knowledge of Chinese medicine techniques, informing them about therapies that effectively alleviate symptoms, such as Taiji spine-strengthening exercises and Eight-Duan Brocade [48]. Patients severely affected by their illnesses during the working day should be instructed in joint home-workplace exercise methods to enhance physical functioning and reduce work-related disabilities [49]. For patients unaccustomed to exercising, it is recommended to use wearable technology to assist in combining home exercises to improve symptoms of discomfort [50]. Patients should also be instructed in alternative therapies, such as aromatherapy, music therapy and cognitive-behavioral therapy, to relieve pain, reduce negative emotions, and improve quality of life [51, 52].

Disease-induced impairment of physical activity can lead to unemployment and increased financial stress [53]. The community can provide employment protection for patients with similar chronic diseases by clarifying their working abilities and offering suitable positions. The 'five-society linkage model'—consisting of the community, medical social workers, community volunteers, community social organizations, and community charitable resources— can be applied to provide health follow-ups [54], establish rehabilitation clubs, and create communication platforms for patients and their families. This ensures continuous support and

**Table 5. Potential solutions to the current problems.**

| Current problems | | Potential solutions |
|---|---|---|
| **Difficulties in diagnosis and treatment** | Medical experience twists and turns | Physicians set up AS teams; timely referral. |
| | The dilemma of pain medication use | Establish an online communication platform for patients. |
| **Effects of disease symptoms** | The diagnosis of the disease caused a bad mood | Pay attention to patients' emotions; cognitive-behavioral therapy; peer support programs. |
| | Pain brings challenges to daily life | Chinese medicine techniques (Taiji and Eight-Duan Brocade); alternative therapies (aromatherapy, music therapy and cognitive-behavioral therapy). |
| | Fatigue affects the quality of life | Exercise training. |
| **Maladjustment of social roles** | Weakened family functioning | Encourage patients to respond positively; |
| | Career planning was disrupted | Home-workplace exercise methods. |
| | There is social isolation | Providers educate the public about the disease. |
| **Lack of support for disease response** | Lack of emotional support | Providers educate the public about the disease; dyadic coping intervention. |
| | Lack of social support | The community provides suitable jobs; conduct health follow-up; start a recovery club. |

attention from social organizations. To address the public's lack of understanding of AS, which hinders patient rehabilitation, the community should fully utilize the linkage model of the five societies to play a pivotal role. First, community managers should conduct periodic disease education programs for residents to enhance health literacy and increase public understanding and acceptance of patients. Second, community volunteers should be encouraged to offer services such as daily care and rehabilitation support. When necessary, patients' financial burdens can be alleviated through public welfare projects, fundraising, and other initiatives. Additionally, medical staff can develop online remote or group personalized intervention programs to expand the coverage of patients who can receive intervention [31].

A summary of potential solutions to the current problems identified in this study is shown in Table 5.

## Strengths and limitations

This study systematically integrates research on the experiences of people with AS at home, in daily life, and at work. Integrating the patient's experiences and perspectives enables healthcare professionals to comprehend the patient's challenges, thereby forming the foundation for developing an individualized intervention plan.

However, this study has some limitations: it lacks a specific analysis of heterogeneity from the perspective of local culture, such as religious beliefs, economic development, and medical care levels. Future research should conduct more in-depth analyses from these perspectives to provide clinical staff with a more comprehensive and reliable basis for managing patients with ankylosing spondylitis.

## Implications and conclusion

AS, a chronic inflammatory autoimmune disease, is difficult to diagnose, which affects the optimal time for treatment. The chronic pain and fatigue caused by AS limit patients' physical, psychological, and social functioning, increase their economic burden, reduce their quality of life, and negatively impact their families and society. Healthcare professionals should leverage

the Internet to enhance public awareness and inclusion of AS, prioritize the mental well-being of diagnosed patients, support active self-management, deliver personalized interventions, and promote rehabilitation and overall quality of life improvement. The community should leverage its strengths to effectively disseminate knowledge about diseases and create an optimal environment for patient rehabilitation.

## Supporting information

**S1 Appendix. Search strategy.**
(DOCX)

**S2 Appendix. PRISMA checklist.**
(DOCX)

**S1 Table. All studies identified in the literature search.**
(DOCX)

**S2 Table. All data extracted from the primary research.**
(DOCX)

## Acknowledgments

We would like to thank Prof. Yunxian Zhou, School of Nursing, Zhejiang University of Traditional Chinese Medicine, for her advice on qualitative research methodology.

## Author Contributions

**Conceptualization:** Yu Li.

**Data curation:** Yu Li, Dongchi Ma.

**Formal analysis:** Yu Li, Dongchi Ma.

**Methodology:** Yu Li, Dongchi Ma.

**Project administration:** Lili Yang.

**Visualization:** Yu Li, Dongchi Ma, Lili Yang.

**Writing – original draft:** Yu Li.

**Writing – review & editing:** Yu Li, Dongchi Ma, Lili Yang.

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
