## [Decision Letter · Decision Letter 0]

30 Jul 2024

PONE-D-24-24187Experiences and perceptions of patients with ankylosing spondylitis: A systematic review and meta-synthesis of qualitative studiesPLOS ONE

Dear Dr. Yang,

Thank you for submitting your manuscript to PLOS ONE. After careful consideration, we feel that it has merit but does not fully meet PLOS ONE’s publication criteria as it currently stands. Therefore, we invite you to submit a revised version of the manuscript that addresses the points raised during the review process.

The manuscript has been written well. Though the message is not new, there has been no meta synthesis done before. Please modify it as per reviewers comments.

We look forward to receiving your revised manuscript.

Kind regards,

Sham Santhanam

Academic Editor

PLOS ONE

Journal Requirements:

Additional Editor Comments:

Congrats on the effort.

Editorial comments:

Reviewer 1

The topic addressed in the study is well known by the rheumatology community and does not add anything new to what is already known. However, the approach to the topic via a meta-synthesis is something new.

I have some comments

1. In the abstract (Page 2, Line11), the word ‘intensive spinal colitis’ needs to be changed as it does not mean anything.

2. Qualitative studies can be very tiring for the reader to interpret and read. Hence it would be wise to include a table summarising the current problems highlighted in the study and their potential solutions as discussed in discussion section.

3. The conclusion needs to be better written. The line (page 13, Line 222) ‘Healthcare professionals should use internet to enhance their knowledge of AS’ does not read well. Please rewrite it to better reflect the study.

4. Most of the problems discussed in the study stem from lack of awareness about the disease in community. It should be discussed and emphasised in the study.

5. Reference 15 and 50 are just protocols and do not have a conclusion. Please remove them and find better references for the text in discussion.

Reviewer 2

Accept

Reviewers' comments:

Reviewer's Responses to Questions

**Comments to the Author**

1. Is the manuscript technically sound, and do the data support the conclusions?

Reviewer #1: Yes

Reviewer #2: Yes

2. Has the statistical analysis been performed appropriately and rigorously? 

Reviewer #1: Yes

Reviewer #2: Yes

3. Have the authors made all data underlying the findings in their manuscript fully available?

Reviewer #1: Yes

Reviewer #2: Yes

4. Is the manuscript presented in an intelligible fashion and written in standard English?

Reviewer #1: Yes

Reviewer #2: Yes

5. Review Comments to the Author

Reviewer #1: The topic addressed in the study is well known by the rheumatology community and does not add anything new to what is already known. However, the approach to the topic via a meta-synthesis is something new.

I have some comments

1. In the abstract (Page 2, Line11), the word ‘intensive spinal colitis’ needs to be changed as it does not mean anything.

2. Qualitative studies can be very tiring for the reader to interpret and read. Hence it would be wise to include a table summarising the current problems highlighted in the study and their potential solutions as discussed in discussion section.

3. The conclusion needs to be better written. The line (page 13, Line 222) ‘Healthcare professionals should use internet to enhance their knowledge of AS’ does not read well. Please rewrite it to better reflect the study.

4. Most of the problems discussed in the study stem from lack of awareness about the disease in community. It should be discussed and emphasised in the study.

5. Reference 15 and 50 are just protocols and do not have a conclusion. Please remove them and find better references for the text in discussion.

Reviewer #2: nil-----------------------------------------------------------------------------------------------------------------------------------------------------------------------------------------------------------------------------------------------------------------------------------------------------------------------------------------------------------------------------------------------

6. PLOS authors have the option to publish the peer review history of their article (what does this mean?). If published, this will include your full peer review and any attached files.

Reviewer #1: **Yes: **Kushagra Gupta

Reviewer #2: **Yes: **Vijaya Prasanna

---

## [Author Response · Author response to Decision Letter 0]

22 Aug 2024

Dear Editors and Reviewers,

We are very grateful for your constructive comments and suggestions regarding our manuscript entitled “Experiences and perceptions of patients with ankylosing spondylitis: A systematic review and meta-synthesis of qualitative studies." Your feedback is invaluable and has been instrumental in improving our manuscript. Below, we provide responses to each comment in turn. In this revised version, changes to our manuscript were all highlighted within the document by using red-colored text. If there are any discrepancies or further questions regarding the manuscript, please do not hesitate to let us know. I look forward to working with you and the reviewers to move this manuscript closer to publication in PLOS ONE.

Looking forward to hearing from you soon.

Yours Sincerely,

Yu Li

Reponse to Editor’s Comments:

Reponse: Thank you for your valuable feedback. We checked again whether our manuscript and file name meet the journal's requirements using the link you shared. According to the template styles, we modified the font on the first page of the manuscript. 

We appreciate your consideration and look forward to your feedback on the revised version.

Reponse: Thank you for your detailed review and valuable suggestions regarding our manuscript. We have rechecked the references to this manuscript and replaced the URL with the DOI number. We have not retracted any of the references, and you can search them using their DOI or PMID numbers. In addition, the original references 15 and 50 have been replaced with new ones. We have also inserted two new references, 45 and 52. Due to the increase in the number of references, the order of some references has changed from the original manuscript. The table below shows the order of references before and after the changes.

Reference number (before) Reference number (after)

[45] [46]

[46] [47]

[47] [48]

[48] [49]

[49] [50]

[50] [51]

[51] [52]

[52] [53]

[53] [54]

Reponse to Reviewer 1’s Comments:

The topic addressed in the study is well known by the rheumatology community and does not add anything new to what is already known. However, the approach to the topic via a meta-synthesis is something new.

Reponse: Thank you for reviewing our work. We appreciate the comments received to further improve the manuscript and have made the necessary changes accordingly.

1. In the abstract (Page 2, Line11), the word ‘intensive spinal colitis’ needs to be changed as it does not mean anything.

Reponse: Thank you for pointing this out. We apologize for the confusion caused by our previous writing. After consideration, we have decided to change 'intensive spinal colitis' to' ankylosing spondylitis'. Please see Page 2, Line 10.

Page 2, Line 10:

The systematic evaluation of relevant qualitative studies on the experiences of patients with ankylosing spondylitis provides a foundation for the clinical development of personalized disease management programs for this patient category.

2. Qualitative studies can be very tiring for the reader to interpret and read. Hence it would be wise to include a table summarising the current problems highlighted in the study and their potential solutions as discussed in discussion section.

Reponse: Thank you for your excellent suggestion. We also think that a tabular summary of current problems and potential solutions is more convenient for readers to read this article. So we created Table 5. Potential solutions to the current problems. In the summary process, we were pleased to find that peer support programs can also help patients positively cope with the disease, so we added this method in the discussion section of the article and Table 5. Please see Page 13, Lines 196-197 and Page 13, Lines 220-221.

Page 13, Lines 196-197:

Moreover, peer support programs can be beneficial during the early stages of coping following diagnosis, aiding patients in better adjusting to life with the disease [45].

Page 13, Lines 220-221:

A summary of potential solutions to the current problems identified in this study is shown in Table 5.

Table 5. Potential solutions to the current problems

current problems potential solutions

Difficulties in diagnosis and treatment Medical experience twists and turns Physicians set up AS teams; 

timely referral.

 The dilemma of pain medication use Establish an online communication platform for patients.

Effects of disease symptoms The diagnosis of the disease caused a bad mood Pay attention to patients' emotions;

cognitive-behavioral therapy;

peer support programs.

 Pain brings challenges to daily life

 Chinese medicine techniques (Taiji 

and Eight-Duan Brocade);

alternative therapies (aromatherapy, music therapy and cognitive-behavioral therapy).

 Fatigue affects the quality of life Exercise training.

Maladjustment of social roles Weakened family functioning Encourage patients to respond positively;

 Career planning was disrupted Home-workplace exercise methods.

 There is social isolation Providers educate the public about the disease.

Lack of support for disease response Lack of emotional support

 Providers educate the public about the disease;

dyadic coping intervention.

 Lack of social support

 The community provides suitable jobs;

conduct health follow-up;

start a recovery club.

3. The conclusion needs to be better written. The line (page 13, Line 222) ‘Healthcare professionals should use internet to enhance their knowledge of AS’ does not read well. Please rewrite it to better reflect the study.

Reponse: Thank you for your valuable feedback. We revised this and other sentences in the conclusion for better reading and understanding. Please see Page 14, Lines 233-237.

Page 14, Lines 233-237:

Healthcare professionals should leverage the Internet to enhance public awareness and inclusion of AS, prioritize the mental well-being of diagnosed patients, support active self-management, deliver personalized interventions, and promote rehabilitation and overall quality of life improvement. The community should leverage its strengths to effectively disseminate knowledge about diseases and create an optimal environment for patient rehabilitation.

4. Most of the problems discussed in the study stem from lack of awareness about the disease in community. It should be discussed and emphasised in the study.

Reponse: Thank you for your insightful feedback on our manuscript. We all agree very strongly with you. In the discussion section, we explained from different angles what troubles people caused by a lack of understanding of diseases and emphasized the importance of raising awareness of diseases. We also have some feasible solutions for the community. Please see Page 12, Lines 178-181 and Page 13, Lines 212-217.

Page 12, Lines 178-181:

In this study, it is evident that both healthcare professionals and the public have an insufficient understanding of AS. A lack of awareness among healthcare professionals hinders accurate disease diagnosis and may compromise the effectiveness of subsequent treatment [11]. Similarly, the public's unfamiliarity with AS can lead to misconceptions about patients, which negatively impacts patient responses to the disease and undermines social harmony.

Page 13, Lines 212-217:

To address the public's lack of understanding of AS, which hinders patient rehabilitation, the community should fully utilize the linkage model of the five societies to play a pivotal role. First, community managers should conduct periodic disease education programs for residents to enhance health literacy and increase public understanding and acceptance of patients. Second, community volunteers should be encouraged to offer services such as daily care and rehabilitation support. When necessary, patients' financial burdens can be alleviated through public welfare projects, fundraising, and other initiatives.

5. Reference 15 and 50 are just protocols and do not have a conclusion. Please remove them and find better references for the text in discussion.

Reponse: Thank you for your detailed feedback. We are very sorry for our inappropriate references to these two articles. We have found new references to replace the original references 15 and 50 and added a new reference 52 on this basis to prove that music therapy, as one of the alternative therapies, can also reduce pain in patients. Due to the increase in the number of references, the order of reference 50 becomes 51. We appreciate your attention to this matter and will make the change in the revised manuscript. Please see Page 16, Lines 300-302 and Page 18, Lines 397-402.

Page 16, Lines 300-302:

15. Hu J, Mao Y, Zhang Y, Ye D, Wen C, Xie Z. Moxibustion for the treatment of ankylosing spondylitis: a systematic review and meta-analysis [J]. Ann Palliat Med, 2020, 9(3): 709-720. doi: 10.21037/apm.2020.02.31. PMID: 32312058.

Page 18, Lines 397-402:

51. Ehde D M, Dillworth T M, Turner J A. Cognitive-behavioral therapy for individuals with chronic pain: efficacy, innovations, and directions for research [J]. American psychologist, 2014, 69(2): 153-166. doi: 10.1037/a0035747. PMID: 24547801.

52. Deng C, Xie Y, Liu Y, Li Y, Xiao Y. Aromatherapy Plus Music Therapy Improve Pain Intensity and Anxiety Scores in Patients With Breast Cancer During Perioperative Periods: A Randomized Controlled Trial [J]. Clinical Breast Cancer, 2022, 22(2): 115-120. doi: 10.1016/j.clbc.2021.05.006. PMID: 34134947.

---

## [Decision Letter · Decision Letter 1]

2 Sep 2024

PONE-D-24-24187R1Experiences and perceptions of patients with ankylosing spondylitis: A systematic review and meta-synthesis of qualitative studiesPLOS ONE

Dear Dr. Yang, Please revise the manuscript as per reviewer comments. Please provide a point by point response and highlight the changes in the manuscript.

**A**Please ensure that your decision is justified on PLOS ONE’s publication criteria and not, for example, on novelty or perceived impact.

We look forward to receiving your revised manuscript.

Kind regards,

Sham Santhanam

Academic Editor

PLOS ONE

Journal Requirements:

Additional Editor Comments:

Please revise the manuscript as per reviewer 1 comments.

Reviewers' comments:

Reviewer's Responses to Questions

**Comments to the Author**

1. If the authors have adequately addressed your comments raised in a previous round of review and you feel that this manuscript is now acceptable for publication, you may indicate that here to bypass the “Comments to the Author” section, enter your conflict of interest statement in the “Confidential to Editor” section, and submit your "Accept" recommendation.

Reviewer #1: All comments have been addressed

2. Is the manuscript technically sound, and do the data support the conclusions?

Reviewer #1: Yes

3. Has the statistical analysis been performed appropriately and rigorously? 

Reviewer #1: Yes

4. Have the authors made all data underlying the findings in their manuscript fully available?

Reviewer #1: Yes

5. Is the manuscript presented in an intelligible fashion and written in standard English?

Reviewer #1: Yes

6. Review Comments to the Author

Reviewer #1: Thank you for making all the suggested corrections.

The phrase 'Intensive spinal colitis' still remains in the main abstract of the article which needs to be changed.

7. PLOS authors have the option to publish the peer review history of their article (what does this mean?). If published, this will include your full peer review and any attached files.

Reviewer #1: **Yes: **Kushagra Gupta

---

## [Author Response · Author response to Decision Letter 1]

2 Sep 2024

Dear Editor and Reviewer,

Thank you very much for giving me this opportunity to revise our manuscript entitled “Experiences and perceptions of patients with ankylosing spondylitis: A systematic review and meta-synthesis of qualitative studies (PONE-D-24-24187R1)” again. According to the review comments, this revision is mainly for the abstract content in the system, and the manuscript content does not need to be modified.

Looking forward to hearing from you soon.

Yours Sincerely,

Yu Li

Reponse to Editor’s Comments:

Reponse: Thank you for your valuable feedback. Through examination, our manuscript meets the criteria of plos one.

Reponse: Thank you for your excellent suggestion. Our manuscript type is a systematic review and meta-synthesis, which contains the results of the study but does not require pilot data.

Reponse to Reviewer 1’s Comments:

1. The phrase 'Intensive spinal colitis' still remains in the main abstract of the article which needs to be changed.

Reponse: Thank you very much for your careful review. In the last revision, we have changed the phrase “intensive spinal colitis” in the manuscript to “ankylosing spondylitis”. However, due to an oversight, we did not check that the term “intensive spinal colitis” was not replaced in the summary of the system. In this submission, we have amended “intensive spinal colitis” to “ankylosing spondylitis” in the system.

---

## [Decision Letter · Decision Letter 2]

25 Sep 2024

Experiences and perceptions of patients with ankylosing spondylitis: A systematic review and meta-synthesis of qualitative studies

PONE-D-24-24187R2

Dear Dr. Yang,

We’re pleased to inform you that your manuscript has been judged scientifically suitable for publication and will be formally accepted for publication once it meets all outstanding technical requirements.

Kind regards,

Sham Santhanam

Academic Editor

PLOS ONE

Additional Editor Comments (optional):

Best wishes

Reviewers' comments:

Reviewer's Responses to Questions

**Comments to the Author**

1. If the authors have adequately addressed your comments raised in a previous round of review and you feel that this manuscript is now acceptable for publication, you may indicate that here to bypass the “Comments to the Author” section, enter your conflict of interest statement in the “Confidential to Editor” section, and submit your "Accept" recommendation.

Reviewer #1: All comments have been addressed

2. Is the manuscript technically sound, and do the data support the conclusions?

Reviewer #1: Yes

3. Has the statistical analysis been performed appropriately and rigorously? 

Reviewer #1: Yes

4. Have the authors made all data underlying the findings in their manuscript fully available?

Reviewer #1: Yes

5. Is the manuscript presented in an intelligible fashion and written in standard English?

Reviewer #1: Yes

6. Review Comments to the Author

Reviewer #1: Thank you for addressing all the queries. There are no further corrections or recommendations pending from my side.

Best of luck with your submission.

7. PLOS authors have the option to publish the peer review history of their article (what does this mean?). If published, this will include your full peer review and any attached files.

Reviewer #1: No

---

## [Editor Report · Acceptance letter]

8 Oct 2024

PONE-D-24-24187R2 

PLOS ONE

Dear Dr. Yang, 

I'm pleased to inform you that your manuscript has been deemed suitable for publication in PLOS ONE. Congratulations! Your manuscript is now being handed over to our production team.

Kind regards, 

on behalf of

Dr. Sham Santhanam 

Academic Editor

PLOS ONE